# Influence of *Auricularia cornea* Polysaccharide Coating on the Stability and Antioxidant Activity of Liposomes Ginsenoside Rh2

**DOI:** 10.3390/foods12213946

**Published:** 2023-10-29

**Authors:** Minghui Wang, Qinyang Li, Shuang Li, Yunzhu Zhao, Xintong Jiang, Sihan He, Junmei Liu

**Affiliations:** 1College of Food Science and Engineering, Jilin Agricultural University, Changchun 130118, China; wangminghui202308@163.com (M.W.); li_shuang2021@163.com (S.L.); zyz18686595857@163.com (Y.Z.); hesihan1@163.com (S.H.); 2Jilin Province Yang Yiduo Technology Co., Ltd., Changchun 130000, China; 3College of Chemistry and Chemical Engineering, South China University of Technology, Guangzhou 510641, China; 202367370151@mail.scut.edu.cn; 4College of Life Sciences and Engineering, Lanzhou University of Technology, Lanzhou 730050, China; 18088611532@163.com

**Keywords:** *Auricularia cornea* polysaccharide, Ginsenosides Rh2, liposomes, gastroenteric fluids stability, antioxidant activity

## Abstract

Liposomes (Lip) are microstructures containing lipid and aqueous phases for encapsulation and delivery of bioactivators. In this study, Ginsenoside Rh2 liposomes (Rh2−Lip) were prepared by a thin-film hydrated ultrasonic binding method. But they are not stable during storage. In addition, Rh2−Lip was wrapped with *Auricultural cornea* polysaccharide (ACP) and Chitosan (CS) as coating materials to improve stability. CS coating was used as a positive control. The particle sizes determined by dynamic light scattering (DLS) showed 183 ± 5.52 nm for liposomes, 197 ± 6.7 nm for *Auricultural cornea* polysaccharide coated liposomes (ACP−Rh2−Lip), and 198 ± 3.5 nm for Chitosan coated liposomes (CS−Rh2−Lip). The polydispersity index (PDI) of all liposomes was less than 0.3. Transmission electron microscopy (TEM) showed that ACP and CS were successfully encapsulated on the liposome surface. In vitro simulations of digestive stability in the gastrointestinal tract showed that ACP−Rh2−Lip and CS−Rh2−Lip were more stable in gastrointestinal fluids compared to Lip. The antioxidant experiment revealed that ACP−Rh2−Lip has greater antioxidant activity than Lip. The purpose of this study was to look into the effects of ACP−Rh2−Lip and to offer a reference for Ginsenoside Rh2 (Rh2) delivery.

## 1. Introduction

Ginseng is a perennial herb of Acanthaceae, which has a history of thousands of years and has a wide range of biological effects [1]. Rh2 is a monomer substance isolated from Ginsenosides, which is a natural active ingredient [2]. It possesses a variety of biological properties including anticancer activity [3], immune enhancement [4], protection of the nervous system [5], anti-inflammatory properties [6], and antioxidant properties [7]. However, since Rh2 is a fat-soluble drug with low solubility in water, and low bioavailability [8], its practical application and development have been limited to some extent. Liposomes, emulsions, and polymer particles have been effectively used as vehicles to address these problems [9]. Among them, liposomes have many beneficial properties such as low toxicity [10], good biocompatibility [11], and target specificity [12].

Liposomes, which are spherical vesicles composed of phospholipids and cholesterol, are promising carriers of hydrophobic and hydrophilic compounds [13]. Liposomes are widely used not only for drug administration but also to protect target products from degradation in food, cosmetics, and agriculture [14]. Liposomes solved the problem of solubility of Red Pigment and efficiently retained the properties of Red Pigment during the biotransport process [15]. Delivery of curcumin using liposomes is a very efficient polyphenol transport system [16]. It has been found that there are many studies related to the delivery of ginsenoside Rh2. Rh2 liposomes studied by Hong et al. [17] were used to prolong circulation in the bloodstream, but the clinical efficacy of liposomes is still limited by the complexity of the tumor microenvironment and the insufficient accumulation at the tumor site. Sun et al. [18] studied ginsenoside Rh2 microsphere hydrogels for the treatment of skin infections. However, in vitro drug release studies showed that Rh2 in the hydrogels exhibited a fast and short release pattern. The in vitro study of ginsenoside Rh2 nanoliposome formulation to enhance the antitumor effect against prostate cancer by Zare-Zardini et al. [19] was used to improve the poor bioavailability of ginsenoside Rh2. However, the use of liposomes in food and beverages is limited because they are susceptible to fusion, aggregation, and leakage of bioactive ingredients during storage. Covering living polymers has been shown to improve the physical stability of liposomes.

In a study, Markus et al. [20] found that ginsenoside Rh2 coupling provided terminal carboxyl groups to dope leaf extracts into nanocomposites to enhance Rh2 antioxidant activity against DPPH, ABTS^+^ radicals as well as aqueous solubility. Natural sugars are often used as stabilizers, such as Chitosan and Alginate. They are used to cover the surface of delivery devices to improve the stability of liposomes [21]. At the same time, the active ingredient polysaccharide has good antioxidant activity. In addition, specific polymer coatings can enable additional functions such as more efficient internalization of liposomes into specific cells and improved ability of the gastrointestinal tract to keep liposomes in place. For example, Chitosan, a cationic polysaccharide present in nature, can interact electrostatically with negatively charged liposomes, with which it mixes, as well as bind anionic polymers to improve the stability of liposomes [22]. Polysaccharide-coated quercetin liposomes, studied by Manuel et al. [23] had an enhanced effect as antioxidants on free radical scavenging activities. Trehalose, which consists of sulfated or non-sulfated galactose, can be used as a thickening agent in liposome suspensions to retard vesicle aggregation and fusion [16]. At the same time, polysaccharide gums can interact with the phosphate group (PO^2−^) of liposomes, reducing membrane fluidity and increasing the accumulation of horizontal phospholipid bilayers [24]. Thus, the binding of polysaccharides to liposomes has significant advantages for improving the bioavailability of drugs. ACP is an edible medicinal fungal polysaccharide with immunity−enhancing, antioxidant, glucose-lowering, and inhibitory properties [25]. It consists of fructose units terminated by terminal glucose groups and can effectively modify liposomes [25]. It is important for improving the stability and bioavailability of liposomes. However, little research has been conducted on polysaccharide-coated liposomes of ACP, an edible mushroom. Therefore, in this paper, we chose to compare CS with ACP to study the effect of polysaccharide coating on Lip stability.

In this study, we prepared CS−Rh2−Lip and ACP−Rh2−Lip. Firstly, the physical properties of polysaccharides-coated liposomes, comprising morphology, PDI, zeta potential, and particle size, were studied by DLS and TEM. Then, the interaction mechanism between polysaccharides and liposomes was studied using Fourier transform infrared spectroscopy (FTIR). In addition, stability and antioxidant activity were determined to assess the protective effect of liposomes against Rh2. This study will contribute to the further application of polysaccharide-coated liposomes from edible medicinal mushrooms in delivering Rh2 and improving its stability.

## 2. Materials and Methods

### 2.1. Materials

Rh2 (purity: 97%) was purchased from Shanghai Yuanye Biotechnology Co., Ltd. (Shanghai, China). *Auricularia cornea* polysaccharide was provided by the team of academician Li Yu from Jilin Agricultural University (Jilin, China). Soy lecithin (from soybean, Mw = 758.06 kDa, 98% purity), cholesterol (95% purity), chitosan (98% purity) mucin (M885077), pepsin (P6322, >3000 U/mg), pancreatic enzymes (P816199), and bile salts were purchased from Shanghai McLean Biochemical Co., Ltd. (Shanghai, China). DPPH (purity ≥ 96%) was purchased from Shanghai Yubo Biotechnology Co., Ltd. (Shanghai, China). The other chemicals used in the experiments were of analytical grade.

### 2.2. Methods

#### 2.2.1. Preparation of Rh2−Lip by Thin Film Hydration [26] Combined with Ultrasonic Method

Before liposome preparation, soy lecithin was dried in a dry nitrogen blower flowing nitrogen (model: DN−12A). Soy lecithin and cholesterol were weighed and mixed with 10 mL of mixed solvent (chloroform:methanol = 6:4). The mixture was introduced into a round-bottomed flask and rotary evaporated at 50 °C for 30 min to remove the residual solvent mixture and to form a uniform film on the wall of the flask. The films were then hydrated with 0.05 M phosphate buffer solution (PBS) by vortexing at 50 °C for 30 min. To obtain nanoscale, the ultrasonic probe was used in an ice bath for 4 min with 5 s pulses on and 5 s pulses off. The generated liposomes were then filtered through a 0.45 μm filtration membrane to obtain a liposome solution, which was transferred to a brown glass vial filled with nitrogen and stored at 4 °C in a light-protected environment for further use.

The encapsulation efficiency (EE) was determined using a UV-visible spectrophotometer (RM-N5000S; Ruiming Instrument Co., Qingdao, China) to observe the maximum absorbance of Rh2 at 544 nm. The Rh2 standard was dissolved in methanol to produce different concentrations of the standard solution, placed in a 50 °C water bath for 1 min to evaporate the methanol, added 0.5 mL of 8% vanillin ethanol test solution, 5 mL of 72% sulfuric acid solution, and heated by thorough mixing in a 60 °C water bath for 15 min, and then immediately used in an ice bath for 10 min and shaken well. At 544 nm, the absorbance was measured. The standard calibration curve of Rh2 was thus obtained (Y = 0.033 × X + 0.0773, r^2^ = 0.999). In the above expression, Y is the absorbance at 544 nm and X is the concentration of Rh2 (μg/mL). The Rh2 content in the liposome supernatant was determined based on the same method described above. The EE was estimated by the following equation [27].
(1)$$\mathrm{EE}\left(\%\right)=\frac{\mathrm{Amount}\;\mathrm{of}\;\mathrm{encapsulated}\;\mathrm{Rh}2}{\mathrm{Total}\;\mathrm{amount}\;\mathrm{of}\;\mathrm{Rh}2}\times 100$$

#### 2.2.2. Design of an Experimental Response Surface

Based on the results of the single-factor test, four significant parameters were determined: membrane material ratio (*w*/*w*) (A), Rh2:soy lecithin (*w*/*w*) (B), hydration temperature (°C) and hydration time (min) (D). EE is one of the most important evaluation metrics for liposomes. RSM was used to assess the effect of the selected parameters on the response EE using a Box-Behnken design (BBD) with four factors and three levels. The range and level of experimental variables studied are shown in Appendix A. Optimal preparation conditions and related studies were obtained through Design Expert 8.0.6.1 for the variables.

#### 2.2.3. Preparation of ACP−Rh2−Lip and CS−Rh2−Lip

The prepared Lip dispersion was added dropwise to the polysaccharide solution of the coating material via a peristaltic pump at a volume ratio of 3:5 and combined with magnetic stirring for 60 min [28]. The titration rate was 2.5 mL/min to mix the polysaccharide solution with the liposome dispersion. The polysaccharide-coated liposome solution was stored in a 4 °C refrigerator for further analysis.

#### 2.2.4. The Vesicle Characterization of Liposomes

The particle size, zeta potential, and PDI distribution of vesicles were measured using DLS using a ZetasizerNano-ZS90 (Shanghai Spectrum Instrument Systems Co., Ltd, Shanghai, China). One mL of liposomes was diluted with 1 mL of PBS and added to the cuvette. Each sample was given two minutes to acclimatize the instrument before testing.

#### 2.2.5. TEM

A G2-F30 transmission electron microscope (TEM, Thermo Fisher Scientific, Waltham, MA, USA) was used to examine the microstructure of liposomes. At room temperature, 10 mL of liposome solution was taken, diluted with water, added dropwise into the copper mesh carbon film, stained with 1% sodium phosphotungstate, dried naturally, and observed and photographed under the transmission electron microscope.

#### 2.2.6. FTIR Spectroscopy

Possible molecular interactions between Rh2, liposomes, and polysaccharides were investigated by FTIR. Analyses were performed using an FTIR spectrometer (Thermo Scientific iS20, Thermo Fisher Scientific, Waltham, MA, USA). All samples were lyophilized in a freeze dryer for at least 24 h before measurement. The lyophilized sample (1 mg) and the chromatographic potassium bromide grade sample (100 mg) were pressed into a homogeneous powder and formed into discs of appropriate thickness. The material was measured in the range of 400–4000 cm^−1^ at room temperature using an FTIR spectrometer. The background was measured on pure potassium bromide disks. The spectra of Rh2 were compared and analyzed with those of Rh2, Rh2−Lip and ACP−Rh2−Lip, CS−Rh2−Lip.

#### 2.2.7. In Vitro Simulation of Gastrointestinal Fluid Digestion Stability

##### Simulate the Preparation of Oral, Gastric, and Intestinal Fluids [29]

To study the effect of Rh2 release from ACP−encapsulated liposomes, we developed an in vitro digestive system model (GIT). The model consisted of an oral phase (simulated salivary fluid, SSF, pH = 6.8), a gastric phase (simulated gastric fluid, SGF, pH = 1.5), and a small intestinal phase (simulated intestinal fluid, SIF, pH = 7.0). The simulated digestion process was completed by shaking in a water bath at 37 °C according to our earlier study. All simulated digests and Rh2−Lip, ACP−Rh2−Lip, and CS−Rh2−Lip should be preheated at 37 °C before mixing.

The following are specific in vitro simulations of digestion operations: Rh2−Lip, ACP−Rh2−Lip, and CS−Rh2−Lip were conjugated to SSF (1:1, *v*/*v*) and given for 10 min to simulate oral digestion. SSF was prepared by dissolving 1.594 g of NaCl, 0.202 g of KCl, and 0.6 g of mucin in 1 L of distilled water. To simulate gastric digestion, orally digested Rh2−Lip, ACP−Rh2−Lip, and CS−Rh2−Lip were mixed with SGF (1:1, *v*/*v*) and taken for 2 h. SGF was prepared by dissolving NaCl (2 g), concentrated HCl (7 mL), and pepsin (3.2 mg/mL) in 1 L of distilled water. To simulate small intestine digestion, gastric juice digestion Rh2−Lip, ACP−Rh2−Lip, and CS−Rh2−Lip is mixed with SIF (1:1, *v*/*v*) and taken for 2 h. K_2_HPO_4_ (6.8 g), NaCl (8.775 g), bile salt (5 g), and pancreatic enzyme (3.2 mg/mL) are dissolved in 1 L of distilled water to create SIF. It is important to remember that before combining with SIF, it is necessary to adjust the pH of stomach digestion to 6.8−7.0.

##### In Vitro Simulated Digestive Stability

The simulated in vitro digestion consisted of three consecutive steps in the mouth, stomach, and intestine. The ratio of Rh2−Lip, ACP−Rh2−Lip, CS−Rh2-Lip, and digestive solution were all 1:1. The ratios were completely mixed. During the experiment, the samples were first digested and then digested again. The digestion reaction was stopped by centrifugation at 5000 rpm for 6 min. The supernatant was taken at the next stage of the digestion experiment. The content of Rh2 in the supernatants of different samples at different stages of digestion was determined.

#### 2.2.8. Antioxidant Activity

##### Determination of DPPH Radical Scavenging Capacity

According to the literature [30], the scavenging rate of DPPH radicals by liposomes was determined. Liposomes (1 mL) were first conjugated to 2 mL, 0.4 M, DPPH ethanol solution and then left to stand for 40 min at 25 °C in the dark. Centrifuge at 3000 rpm for 10 min using a high−speed cryo−centrifuge. Finally, the absorbance of the supernatant was measured spectrophotometrically at 517 nm, using Vc as a positive control. The control sample instead of liposome samples, was created using ultrapure water (A_blank_). Instead of using a DPPH solution to prepare the control sample (A_control_), ethanol was used. The following formula was used to compute the DPPH radical scavenging activity:(2)$$\mathrm{DPPH}\;\mathrm{scavenging}\left(\%\right)=\frac{1-\left(\mathrm{Asample}-\mathrm{Acontrol}\right)\;}{\mathrm{Ablank}}\times 100$$

##### Determination of Hydroxyl Radical Scavenging Capacity

The hydroxyl radical scavenging rate was calculated using the method described above [30]. Liposome samples (1 mL) were treated for 30 min in the dark with salicylic acid ethanol solution (9 M), FeCl_2_ (9 M), and H_2_O_2_ (1 mL, 8.8 M). Using a spectrophotometer, the reaction mixture’s absorbance A_1_ was determined at 510 nm. Blank samples were prepared with ultrapure water (A_0_) instead of nanoliposome samples. The salicylic acid ethanol solution (A_2_) was omitted from the preparation of the control sample (A_2_). The following formula was used to compute the rate of hydroxyl radical scavenging.
(3)$$\mathrm{The}\;\mathrm{hydroxyl}\;\mathrm{scavenging}\left(\%\right)=\frac{1-\left(A_{1}-A_{2}\right)\;}{A_{0}}\times 100$$

##### Reducing Power

Add 1 mL of sample solution (1 g/L), 2.5 mL PBS, and 1 mL of potassium ferricyanide (1%) to the tube, mix well with full shaking, and react in a water bath at 50 °C for 20 min. After rapid cooling with running water, 2.5 mL of trichloroacetic acid (10%, *w*/*v*) was added, and finally, 1 mL of ferric chloride (0.1%, *w*/*v*) was added to mix, reacted at room temperature for 10 min, and the OD value was measured at 700 nm, and the blank tube was replaced by distilled water instead of polysaccharide solution [30].

#### 2.2.9. Statistical Analysis

The mean and standard deviation (mean SD) of all data were calculated. Using the SPSS 26.0 program, all statistical comparisons were made between all of the results. To assess significant sample differences, a one-way analysis of variance (ANOVA) was utilized. A *p*-value of less than 0.05 (*p* < 0.05) is used to define statistical significance. Each experiment is performed in triplicate, and each sample is freshly made.

## 3. Results

### 3.1. Preparation of Rh2-Lip

The results of the response surface optimization experiments with membrane material ratio (A), Rh2:soy lecithin (B), hydration temperature (C), and hydration time (D) as the independent variables and liposome EE as the response value are shown in Appendix A. A multivariate regression was fitted to the data in Appendix A to obtain a quadratic polynomial regression model for the EE and independent factors:Y = 43.87117 + 0.62063 × A + 47.01276 × B + 1.1328 × C + 0.76924 × D + 3.56463 × A × B + 0.010667 × A × C + 0.044667 × A × D − 0.1394 × B × C + 0.16019 × B × D − 1.38414 × A^2^ − 462.59144 × B^2^ − 0.01174 × C^2^ − 0.010604 × D^2^

The results of the ANOVA for each factor are shown in Appendix A. It can be seen that the quadratic terms for membrane material ratio, Rh2:soy lecithin, hydration temperature, and hydration time were significant. The F-value can be used to analyze the extent to which the EE of Rh2−Lip is affected by a single factor. In Appendix A, A > D > B > C, the degree of influence of these four factors on liposome EE was membrane material ratio > hydration time > Rh2:soy lecithin > hydration temperature. The *p*-values of the quadratic terms of the four terms are less than 0.05, indicating that the quadratic terms of the four terms have a significant effect on the EE of Rh2−Lip. The quadratic model of the regression model was highly significant (*p* < 0.01), indicating that the model was a good fit for the experiment within the experimental range, with a moderate model choice and high reliability of the experimental results. Therefore, it is feasible to utilize the model to predict the EE of Rh2−Lip.

By depicting the response surfaces in three dimensions, as in Figure 1, the interaction of four independent factors (A−D) on EE was demonstrated. The EE of Rh2-Lip is correlated with the above four factors, and the effect of each factor on the EE can be maximized. The quadratic regression model was analyzed. Based on the analysis of the Box−Behnken design model, the optimized process parameters of Rh2−Lip were as follows: membrane material ratio 1:1, Rh2: soy lecithin 1:15, hydration temperature 50 °C, hydration time 40 min. Under these conditions, five sets of parallel experiments were conducted. The average EE of the prepared Rh2−Lip was 89.526%, which was close to the predicted value of 89.5005%, proving that the preparation process optimized by the response surface methodology has high accuracy and reference value.

### 3.2. Preparation of ACP-Rh2-Lip and CS-Rh2-Lip

Increasing concentrations of antioxidants may induce pro-oxidation of lipids, leading to liposome instability [31]. Therefore, in this study, the concentration of ACP was 2.5 mg/mL and the concentration of CS was 1.0 mg/mL, at which time the EE and storage stability of liposomes were relatively high. In this study, three different concentrations of polysaccharides were tested to coat liposomes. The prepared liposome dispersion was added dropwise to the polysaccharide solution of the coating material under magnetic stirring for 60 min to give the polysaccharide enough time to contact, react, and adsorb on the liposome surface. As can be seen from Figure 2, using CS−Rh2−Lip as a positive control, the optimal EE of ACP−Rh2−Lip was 91.2% when the concentration of ACP was 2.5 mg/mL, which was significantly different from other concentrations of ACP−Rh2−Lip. When the CS concentration was 1 mg/mL, the optimal EE of CS−Rh2−Lip was 90.15%, which was significantly different from other concentrations of CS−Rh2−Lip.

### 3.3. Particle Size, Zeta Potential, and PDI

The physical properties of liposomes were characterized using the DLS technique [32]. Particle size is a fundamental parameter of liposomes; smaller particle sizes can create repulsive forces between liposome vesicles and outweigh the attractive forces, helping to stabilize the suspension of liposomes [33]. The particle sizes of liposomes with different encapsulation rates are shown in Figure 3A. The coating of polysaccharides increases the particle size due to the increased thickness and curvature of the nanoliposome vesicles.

PDI is a metric used to assess the homogeneity of particle size in suspensions. In general, the smaller the PDI, the better the particle dispersion scattering. As shown in Figure 3A, the PDI values of Empty−Rh2, Rh2−Lip, ACP−Rh2−Lip, and CS−Rh2−Lip were elevated but remained below 0.3, which is consistent with the findings of Xue et al. [34], indicating uniform dispersion.

The zeta potential is a parameter representing the surface charge of the particles and correlates with the stability of colloidal suspensions [35]. A larger absolute value of the zeta potential indicates a higher surface charge on the vesicles of the nanoliposomes, which increases the degree of repulsion between the vesicles and protects the nanoliposomes from aggregation [36]. Figure 3B shows that the zeta potential of Lip was −21.9 ± 0.61 mV due to the presence of negatively charged phosphatidic acid in soy lecithin. After the addition of differently charged polysaccharides, the values of CS−Rh2−Lip and ACP−Rh2−Lip potentials were increased to 22.37 ± 0.65 mV and 28.46 ± 1.27 mV, respectively. The high absolute values indicate higher stability of the nanoliposomes. This observation is mainly attributed to the different surface charges of the polysaccharides themselves.

### 3.4. TEM

TEM is further used to study the effect of polysaccharide coating on the morphology of liposomes. To understand the encapsulation of liposomes for ginsenoside Rh2, TEM was utilized to observe the microscopic morphology of liposomes. As shown in Figure 4, Empty−Lip, Rh2−Lip, ACP−Rh2−Lip, and CS−Rh2−Lip all showed regular spherical shapes, and the particle sizes of ACP−Rh2−Lip and CS−Rh2−Lip were slightly increased compared with those of Lip because the binding of ACP and CS to the liposomes increased the particle size of the liposomes. Meanwhile, a layer of dark material with a lighter contour can be observed in Figure 4A,B, which may be the polysaccharides bound to the surface of the liposomes [37]. The particle size of liposomes observed under TEM was larger compared with that measured by nano-particle sizing, which may be because the samples need to be struck by high-voltage electrons during transmission electron microscopy, and the samples may undergo a certain degree of swelling and disintegration [37], and this larger particle size tendency was observed in Lip, ACP−Rh2−Lip, and CS−Rh2−Lip. This tendency of larger particle size is reflected in the TEM images of Lip, ACP−Rh2−Lip, and CS−Rh2−Lip. These results confirm that polysaccharides are encapsulated on the surface of nanoliposomes and the resulting higher surface charge reduces the aggregation of liposomes.

### 3.5. FTIR Spectroscopy

Fourier infrared spectroscopy was used to determine the structure of the two liposomes, based on the bandwidth and frequency of whether Rh2 was successfully encapsulated within the liposomes and whether the polysaccharide was successfully bound to the liposomes. The results are shown in Figure 5. In the infrared spectrum of Rh2, the peak at 1018 cm^−1^ represents the stretching vibration of C−N, a characteristic peak of ginsenoside Rh2 [20], and is shown at 865 cm^−1^ in Lip, whereas this peak does not appear in Empty−Lip, suggesting that ginsenoside Rh2 was successfully encapsulated within the liposome. Markus et al. [20] suggested that methylene (−CH_2_−) is the absorption peak of polysaccharide substances in the infrared spectrum of ACP at 2884 cm^−1^. It was significantly shifted to the right in ACP−Rh2−Lip, indicating that the ACP was successfully bound to liposomes. ACP−Rh2−Lip reduced the absorption of the characteristic C−O peak at 1054 cm^−1^, which may be caused by the formation of hydrogen bonds between the C−O of the liposomes and the polysaccharide [38]. In the infrared spectra of CS, 1668 cm^−1^ and 1597 cm^−1^ are carbonyl (C=O) and protonated amino (NH^3+^) vibrations in CONHR, respectively [39]. The characteristic peak at 1597 cm^−1^ in chitosan moved to 1626 cm^−1^ in CS−Rh2−Lip, indicating that chitosan was successfully bound to liposomes. In addition, 1160 cm^−1^ in Lip, which is the characteristic moiety of the lipid molecule, phosphate (PO^2−^) [24], moved to 1178 cm^−1^ in ACP−Rh2−Lip and 1171 cm^−1^ in CS−Rh2−Lip, suggesting that liposomes lead to dehydration of the phosphate moiety. In contrast, the characteristic peak of the polysaccharide, the protonated amino group NH^3+^, also shifted toward higher frequencies in the ACP−Rh2−Lip and CS−Rh2−Lip IR spectra, suggesting that most of the hydrogen bonding of the phosphate group was disrupted due to electrostatic interactions with the amino group.

### 3.6. In Vitro Simulated Gastroenteric Fluid Digestive Stability

Figure 6 shows that after oral digestion, Rh2−Lip is relatively well stabilized and the presence of Rh2 is rarely found. After entering the gastric fluid, the stability of liposomes varied with the change in pH value. After immersion in gastric juice for 90 min, the leakage rate of Rh2−Lip reached 50%, while the leakage rates of ACP−Rh2−Lip and CS−Rh2−Lip were much lower than those of Rh2-Lip. This is consistent with the findings of Zhang et al. [40]. The reason for this may be that changes in pH disrupt the structure of the phospholipid bilayer. The neutral gastric environment and gastric electrolyte solution system altered the stability and conformation of polysaccharides, leading to partial leakage of Rh2. However, due to the coating effect of polysaccharides, the surface of liposomes was coated so that the phospholipid bilayer of polysaccharide-coated liposomes was not damaged. It can be seen that the polysaccharide coating has a significant protective effect on Rh2−Lip. Entering the intestinal fluid stage, probably because trypsin hydrolyzed part of the phosphatidylcholine [41], the leakage rate of Rh2−Lip increased extremely fast. As can be seen from the figure, the release of polysaccharide−coated liposomes in the intestinal fluid was significantly slower, suggesting that polysaccharides have a certain protective effect on liposomes.

### 3.7. Antioxidant Activity

Glycosidic bond breaking of ACP produces low molecular weight polysaccharides, which contain high amounts of reduced free hydroxyl groups, thus polysaccharides have antioxidant capacity. DPPH radical scavenging, hydroxyl radical scavenging, and Fe^3+^ reducing capacity can be used to measure the level of antioxidants [42]. Figure 7 shows the results of DPPH radical scavenging rate, hydroxyl radical scavenging rate, and Fe^3+^ reducing the capacity of Rh2−Lip, ACP−Rh2−Lip, and CS−Rh2−Lip at a concentration of 1 mg/mL using Vc as positive control. As can be seen from Figure 7A, polysaccharide coating has a higher DPPH radical scavenging capacity. The high degree of binding of polysaccharides to the surface of liposomes is believed to significantly enhance antioxidant activity. Interestingly, Figure 7B shows that ACP−Rh2−Lip significantly decreased the hydroxyl radical scavenging rate of liposomes, while CS−Rh2−Lip did not change the hydroxyl radical scavenging rate of liposomes, which was mainly related to the polyhydroxy groups in the molecular chains of ACP-Rh2-Lip and CS-Rh2-Lip. The iron reducing power of Rh2, Rh2−Lip, ACP−Rh2−Lip and CS− Rh2−Lip, Rh2−Lip, ACP−Rh2−Lip, and CS−Rh2−Lip had iron reducing power of 0.2905 ± 0.01, 0.3705 ± 0.03, 1.2305 ± 0.05, and 0.979 ± 0.03, respectively. The polysaccharide coatings significantly improved the antioxidant capacity of Rh2−Lip. On the one hand, the thick network covering the liposome surface prevented the interaction of foreign substances with the lipid-water interface. On the other hand, the coverage of polysaccharides tightly organized the lipid acyl chains and hindered the penetration of pro-oxidant substances [16]. The CS contained more reducing hydroxyl groups, which, when combined with reactive oxygen species in the lipid peroxidation chain reaction, prevented the peroxidation process and improved the antioxidant capacity of liposomes [25]. The results suggest that polysaccharide-coated liposomes are a successful method to reduce oxidative stress in liposomes.

## 4. Conclusions

The effect of ACP on the stability and antioxidant activity of Rh2−Lip was revealed in this investigation. First, ACP has a stronger inhibitory effect on the aggregation of Rh2−Lip. Second, additional hydrogen bonds between ACP and Rh2−Lip were formed. At the same time, ACP can compact the inner chain and membrane of lipid molecules, and Rh2−Lip’s stability is thus improved. Furthermore, the intimate interaction between polysaccharides and liposomes significantly inhibited the oxidation reaction in the membrane. The findings of this study could give a scientific foundation for anionic polysaccharide-coated liposomes to be used as an effective carrier for Rh2. Future studies will concentrate on the use of polysaccharide-encapsulated Rh2−Lip in food systems, such as functional confectionery and beverages.

## Figures and Tables

**Figure 1 foods-12-03946-f001:**
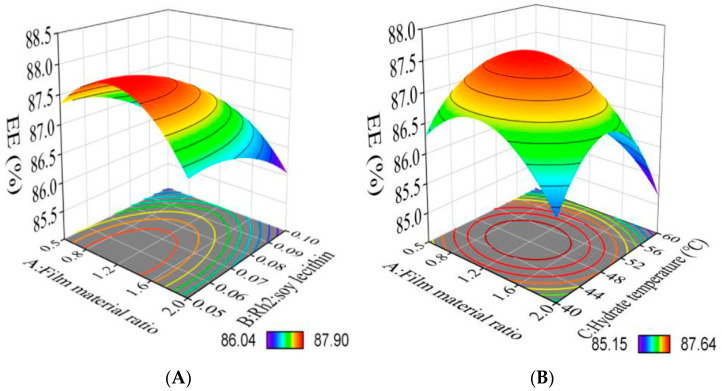
Response surface model plot showing the effects of ratio of film to material ratio to Rh2:Soy lecithin (**A**), ratio of film to material ratio to hydrate temperature (**B**), ratio of film to material ratio to hydrate time (**C**), ratio of Rh2:Soy lecithin to hydrate temperature (**D**), ratio of Rh2:Soy lecithin to hydrate time (**E**), and ratio of Rh2:Soy lecithin to hydrate time (**F**) on encapsulation efficiency.

**Figure 2 foods-12-03946-f002:**
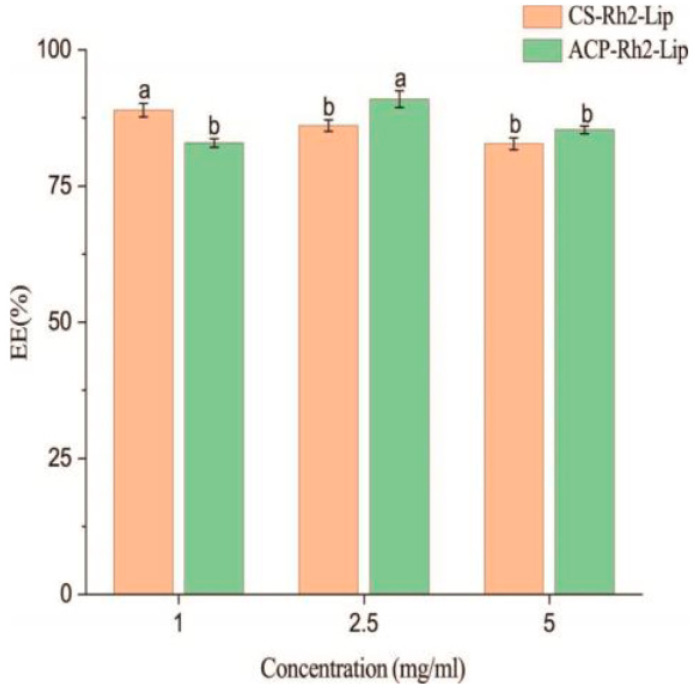
The EE of different concentrations of ACP−Rh2−Lip and CS−Rh2−Lip. Different lowercase letters represent significant differences (*p* < 0.05).

**Figure 3 foods-12-03946-f003:**
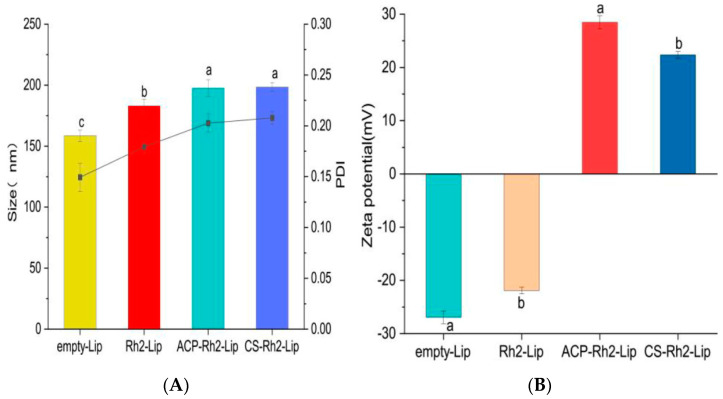
Physical properties of Rh2−Lip, ACP−Rh2−Lip and CS−Rh2−Lip. (**A**) Size and PDI, (**B**) Zeta potential. Different lowercase letters represent significant differences (*p* < 0.05).

**Figure 4 foods-12-03946-f004:**
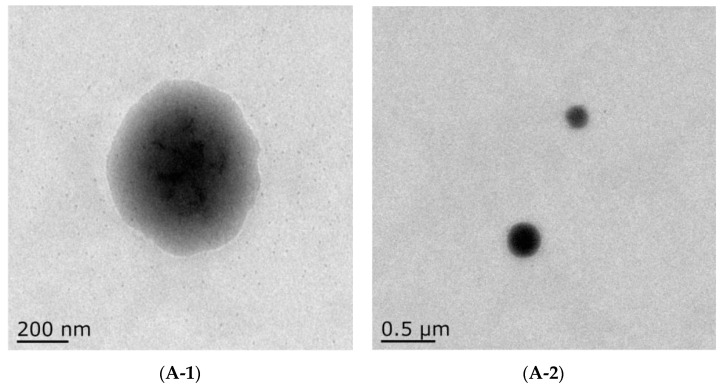
TEM of different liposomes. (**A**) Empty−Lip, (**B**) Rh2−Lip, (**C**) ACP−Rh2−Lip, (**D**) CS−Rh2−Lip.

**Figure 5 foods-12-03946-f005:**
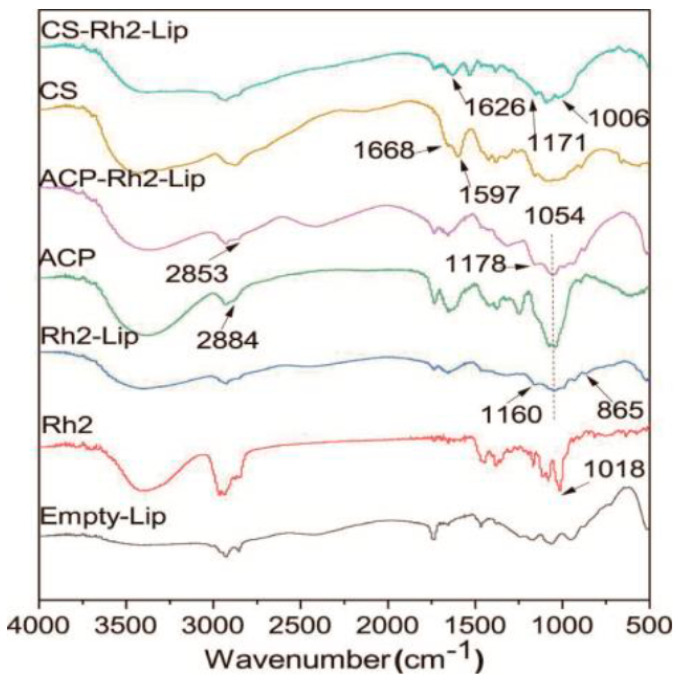
Fourier transform infrared spectra of different liposomes.

**Figure 6 foods-12-03946-f006:**
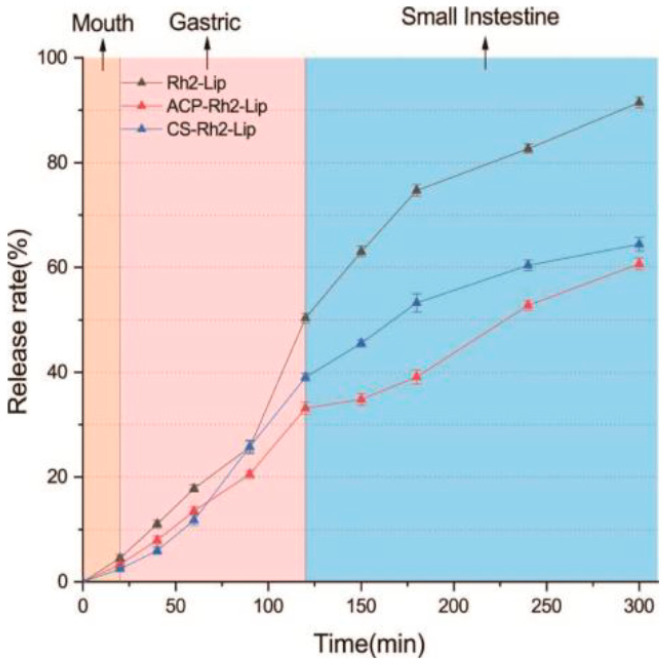
Digestion stability of Rh2−Lip, ACP−Rh2−Lip, and CS−Rh2−Lip simulated gastrointestinal fluid.

**Figure 7 foods-12-03946-f007:**
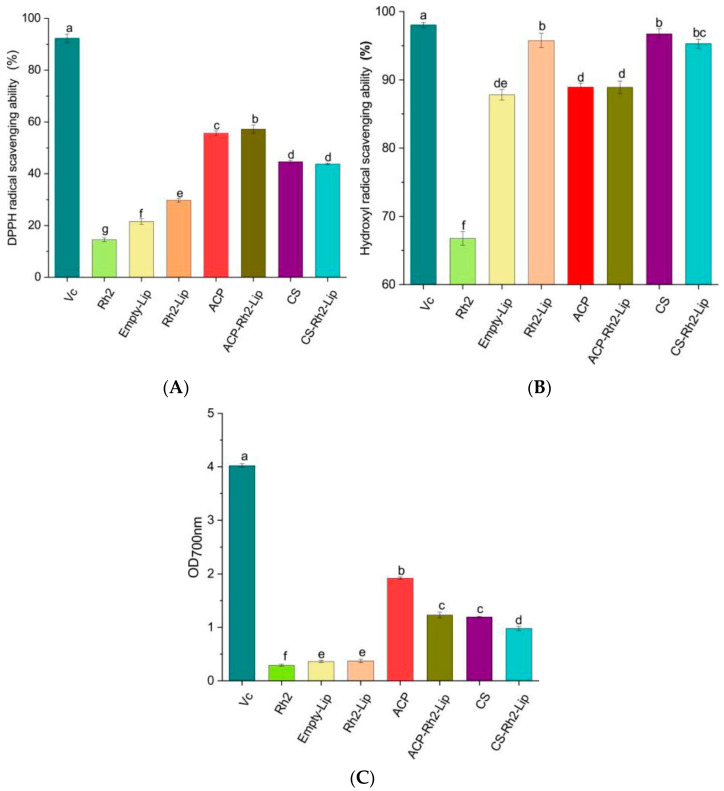
Antioxidant activity of different liposomes, (**A**) DPPH radical scavenging activity, (**B**) Hydroxyl radical scavenging activity, (**C**) Reduction ability of Fe^3+^. Different lowercase letters represent significant differences (*p* < 0.05).

## Data Availability

The data presented in this study are available upon request from the corresponding author.

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
