# Peer review of "Influence of *Auricularia cornea* Polysaccharide Coating on the Stability and Antioxidant Activity of Liposomes Ginsenoside Rh2"

_foods, 2023, doi:10.3390/foods12213946_

Round 1
Reviewer 1 Report
-The authors not declared of mixed solvent to dissolved the soybean lecithin and cholesterol. Usually using Chloroform but chloroform is toxic.
During preparation of pro liposome the authors not using Nitrogen gas to prevent oxidation. Because soy bean lecithin contain phosphatidic acid which easy oxidation and make the un stability of liposome solution.
-During determination of entrapment Efficiency ( EE) the author the authors using centrifuged at 3000 rpm. It is can be separated ?. Because liposome is colloidal solution and must be separated with ultracentrifuge using 30000 RPM to separated the supernatant solution. Or can be using centrifuge filter if they want using only 3000 RPM.
Author Response
|
Response to Reviewer 1 Comments
|
||
|
1. Summary |
|
|
|
Thank you for your constructive comments concerning our article. These comments are valuable and helpful for improving our article. All the authors have seriously discussed about all these comments and revised the manuscript. |
||
|
2. Questions for General Evaluation |
Reviewer’s Evaluation |
Response and Revisions |
|
Does the introduction provide sufficient background and include all relevant references? |
Yes |
|
|
Are all the cited references relevant to the research? |
Yes |
|
|
Is the research design appropriate? |
Yes |
|
|
Are the methods adequately described? |
Can be improved |
We made complete changes, which are detailed below in this document. |
|
Are the results clearly presented? |
Yes |
|
|
Are the conclusions supported by the results? |
Yes |
|
|
3. Point-by-point response to Comments and Suggestions for Authors |
||
|
Comments 1: The authors not declared of mixed solvent to dissolved the soybean lecithin and cholesterol. Usually using Chloroform but chloroform is toxic. |
||
|
Response 1: Thank you for pointing this out. We agree with this comment. Therefore,we have added the mixed solvent component to the revised manuscript, and in the rotary evaporation session for preparation of the liposomes, we have evaporated the mixed solvent for 30 min to ensure the removal of all residual mixed solvents. Soy lecithin and cholesterol were weighed and mixed with 10 ml of mixed solvent (chloroform:methanol=6:4). The mixture was introduced into a round-bottomed flask and rotary evaporated at 50°C for 30 min to remove the residual solvent mixture and form a uniform film on the bottle wall. (Lines 99-102) References Peng, Li, Kuo, Chen, & Lili, et al. Preparation of modified graphene oxide/polyethyleneimine film with enhanced hydrogen barrier properties by reactive layer-by-layer self-assembly. Compos B Eng, 2019. https://doi.org/10.1016/j.compositesb.2019.02.058 |
||
|
Comments 2: During preparation of pro liposome the authors not using Nitrogen gas to prevent oxidation. Because soy bean lecithin contain phosphatidic acid which easy oxidation and make the un stability of liposome solution. |
||
|
Response 2: Thank you for your careful review. Soy lecithin and cholesterol were dried under flowing nitrogen gas before the beginning of the experiment, and the instrument used was the dry nitrogen blower, model: DN-12A. |
||
|
Comments 3: During determination of entrapment Efficiency ( EE) the author the authors using centrifuged at 3000 rpm. It is can be separated? Because liposome is colloidal solution and must be separated with ultracentrifuge using 30000 RPM to separated the supernatant solution. Or can be using centrifuge filter if they want using only 3000 RPM. |
||
|
Response 3: Thank you for pointing this out. We agree with this comment. We have filtered the liposomes using a 0.45um centrifugal filter when the liposome preparation was completed, and in the process of determining the encapsulation rate, centrifugation at 3,000 rpm was used here in order to completely remove the residual core material ginsenoside Rh2. |
||

Reviewer 2 Report
Targeted delivery systems based on liposomes are, one might say, classic; they deliver almost any substrate. However, in the work of Junmei Liu et al., the main substrate was not drugs, but activators of biological processes. What is considered the next stage in the development of personalized medicine. The authors encountered the problem of stabilizing limoposomes and the article was devoted to studying this issue. I liked the work. It is comprehensive and complete. Links to sources are relevant and correct, which is not unimportant: only 1 out of 30 links refers to publications older than 5 years. This indicates the relevance and popularity of the topic.There is only one technical note about this work.
Figure 4 does not have enough statistics, that is, TEM images at various magnifications, as well as diagrams of the size distribution of nanoparticles of vesicles.
Author Response
|
Response to Reviewer 2 Comments
|
||
|
1. Summary |
|
|
|
Thank you for your constructive comments concerning our article. These comments are valuable and helpful for improving our article. All the authors have seriously discussed about all these comments and revised the manuscript.
|
||
|
2. Questions for General Evaluation |
Reviewer’s Evaluation |
Response and Revisions |
|
Does the introduction provide sufficient background and include all relevant references? |
Can be improved |
We made complete changes, which are detailed below in this document. |
|
Are all the cited references relevant to the research? |
Yes |
|
|
Is the research design appropriate? |
Yes |
|
|
Are the methods adequately described? |
Yes |
|
|
Are the results clearly presented? |
Can be improved |
We made complete changes, which are detailed below in this document. |
|
Are the conclusions supported by the results? |
Yes |
|
|
3. Point-by-point response to Comments and Suggestions for Authors |
||
|
Comments 1: Targeted delivery systems based on liposomes are, one might say, classic; they deliver almost any substrate. However, in the work of Junmei Liu et al., the main substrate was not drugs, but activators of biological processes. What is considered the next stage in the development of personalized medicine. The authors encountered the problem of stabilizing limoposomes and the article was devoted to studying this issue. I liked the work. It is comprehensive and complete. Links to sources are relevant and correct, which is not unimportant: only 1 out of 30 links refers to publications older than 5 years. This indicates the relevance and popularity of the topic.
There is only one technical note about this work.
Figure 4 does not have enough statistics, that is, TEM images at various magnifications, as well as diagrams of the size distribution of nanoparticles of vesicles.
|
||
|
Response 1: Thank you for pointing this out. We agree with this comment. Therefore, we have manuscript uploaded transmission electron microscopy images at different magnifications. Empty-Lip is a transmission electron microscopy image at 200 nm and 0.5 μm magnification, Rh2-Lip and ACP-Rh2-Lip are transmission electron microscopy images at 100 nm and 0.5 μm magnification, and CS-Rh2-Lip is a transmission electron microscopy image at 0.5 μm and 1.0 μm magnification. |
||

Reviewer 3 Report
Some parts of the manuscript are incomprehensible and too much gaps are present.
Auricularia Cornea has to be written in italics.
Line 34: what does it mean that ginsenoside is a monomer compound?! Describe the structure.
Line 54: trehalose is not a polysaccharide. It is used as a cryoprotectant during freeze-drying process, not as a stabilizer for liposome stability.
Line 59: chitosan is not ANIONIC.
Lines 64 – 70 are out of scope and used to cite previous works. On the contrary, little information about Auricularia Cornea polysaccharide was provided. As an example, is it charged? In the results you stated that liposomes coated with the polysaccharide are positively charged.
Line 74: glucose is not a group.
Section 2.2.1
What do you mean with “ginsenoside Rh2 solution” if ginsenoside is not soluble in water? Why didn’t you olubilize ginsenoside into the organic solvent along with phospholipids?
Section 2.2.2: and cholesterol??
Section 2.2.3: how did you solubilize the coating material?
Section 2.2.4: it has no sense.
RESULTS: The discussion with the literature is nonexistent.
Section 3.5: you must also record the spectra of physical mixtures of raw materials.
In figure 6 you reported the release of the drug. Thus, you can’t talk about liposome stability in the results. Provide stability results.
Section 3.7: provide the antioxidant capacities of the raw materials.
Poor.
Author Response
|
Response to Reviewer 3 Comments
|
||
|
1. Summary |
|
|
|
Thank you for your constructive comments on our articles. We apologize for the errors in this manuscript and the inconvenience it caused you to read it. The manuscript has been thoroughly revised and edited by native speakers. Thank you very much for your helpful comments. These comments are valuable and helpful in improving our articles. All authors discussed all these comments carefully and revised the manuscript. |
||
|
2. Questions for General Evaluation |
Reviewer’s Evaluation |
Response and Revisions |
|
Does the introduction provide sufficient background and include all relevant references? |
Must be improved |
We made complete changes, which are detailed below in this document. |
|
Are all the cited references relevant to the research? |
Must be improved |
We made complete changes, which are detailed below in this document. |
|
Is the research design appropriate? |
Must be improved |
We made complete changes, which are detailed below in this document. |
|
Are the methods adequately described? |
Must be improved |
We made complete changes, which are detailed below in this document. |
|
Are the results clearly presented? |
Must be improved |
We made complete changes, which are detailed below in this document. |
|
Are the conclusions supported by the results? |
Must be improved |
We made complete changes, which are detailed below in this document. |
|
3. Point-by-point response to Comments and Suggestions for Authors |
||
|
Comments 1: Some parts of the manuscript are incomprehensible and too much gaps are present. |
||
|
Response 1: Thank you for your valuable comments, the manuscript has been revised and corrected, and formatting changes have been made to remove the blank portions of the manuscript. |
||
|
Comments 2: Auricularia Cornea has to be written in italics. |
||
|
Response 2: Thank you for your careful review, we have revised all of the Auricularia Cornea for use in italics. |
||
|
Comments 3: Line 34: what does it mean that ginsenoside is a monomer compound?! Describe the structure. |
||
|
Response 3: Thank you for your careful review, Ginsenoside Rh2, as a natural active ingredient, is a single compound isolated and extracted from ginseng. We have made the changes. Rh2 is a monomer substance isolated from Ginsenosides, which is a natural active ingredient. (Line 34) References Yan H, Jin H, Fu Y, Yin Z, Yin C. Production of Rare Ginsenosides Rg3 and Rh2 by Endophytic Bacteria from Panax ginseng. J Agric Food Chem. 2019;67(31):8493-8499. https://doi.org/10.1021/acs.jafc.9b03159 |
||
|
Comments 4: Line 54: trehalose is not a polysaccharide. It is used as a cryoprotectant during freeze-drying process, not as a stabilizer for liposome stability. |
||
|
Response 4: Thank you for your careful review. Agreed. So we have made changes. Natural sugars are often used as stabilizers, such as chitosan and alginate. They are used to cover the surface of delivery devices to improve the stability of liposomes. (Lines 53-54) References Yuba E, Kado Y, Kasho N, Harada A. Cationic lipid potentiated the adjuvanticity of polysaccharide derivative-modified liposome vaccines. J Control Release. 2022;S0168-3659(22)00686-1. https://doi.org/10.1016/j.jconrel.2022.10.016 |
||
|
Comments 5: Line 59: chitosan is not ANIONIC. |
||
|
Response 5: Thank you for your valuable comments, we have corrected chitosan as a naturally occurring cationic polysaccharide in the revised manuscript, and the related error has been corrected. Chitosan, a cationic polysaccharide present in nature. (Line 59) |
||
|
Comments 6: Lines 64 – 70 are out of scope and used to cite previous works. On the contrary, little information about Auricularia Cornea polysaccharide was provided. As an example, is it charged? In the results you stated that liposomes coated with the polysaccharide are positively charged. |
||
|
Response 6: Thank you for your valuable input. The citation of the previous example is to express that polysaccharide coating action can improve the stability and targeting of liposome delivery system and the way liposomes form a coating action with polysaccharides, and that during the binding process of alginate to polysaccharides, the polysaccharide gums can interact with the phosphate groups (PO43-) of liposomes, and that there is a corresponding change in this paper when yucca polysaccharides are coated with liposomes. So cited. ACP was not changed, but rather the liposomes themselves were shown to be negatively charged by zeta potential measurements, and since the polysaccharides were positively charged, according to the FTIR it can be seen that due to hydrogen bonding, the liposomes bind to the polysaccharides. |
||
|
Comments 7: Line 74: glucose is not a group. |
||
|
Response 7: Thanks for the heads up, we agree with you.ACP consists of fructose terminated by terminal glucose. Not that glucose is not a group. It consists of fructose units terminated by terminal glucose groups and can effectively modify liposomes. (Lines 72-73) References Wang J, Liu B, Qi Y, et al. Impact of Auricularia cornea var. Li polysaccharides on the physicochemical, textual, flavor, and antioxidant properties of set yogurt. Int J Biol Macromol. 2022;206:148-158. https://doi.org/10.1016/j.ijbiomac.2022.02.141 |
||
|
Comments 8: Section 2.2.1: What do you mean with “ginsenoside Rh2 solution” if ginsenoside is not soluble in water? Why didn’t you olubilize ginsenoside into the organic solvent along with phospholipids? |
||
|
Response 8: Thank you for your valuable suggestion. Ginsenoside Rh2 is an alcohol-soluble substance, readily soluble in ethanol; soluble in methanol and almost insoluble in water. The ginsenoside Rh2 solution in the text refers to the liquid containing ginsenoside Rh2 in a phosphate buffer solution. Because ginsenoside Rh2 is an alcohol-soluble substance, it is dissolved when put into organic solvents together with soy lecithin, causing losses during rotary evaporation. Ginsenoside Rh2 and 50 ml of phosphate buffer solution were added to a round-bottomed flask, swirled, and hydrated for 30 min at a certain temperature. (Lines 103-104) References Mathiyalagan R, Wang C, Kim YJ, et al. Preparation of Polyethylene Glycol-Ginsenoside Rh1 and Rh2 Conjugates and Their Efficacy against Lung Cancer and Inflammation. Molecules. 2019;24(23):4367. https://doi.org/10.3390/molecules24234367 |
||
|
Comments 9: Section 2.2.2: and cholesterol?? |
||
|
Response 9: Thank you for your question. It is the further optimization of single factor for liposome preparation including membrane material ratio (w/w) ( A), Rh2: Soy lecithin (w/w) ( B), hydration temperature (℃) and hydration time (min) (D). to give the liposomes a higher encapsulation rate. |
||
|
Comments 10: Section 2.2.3: how did you solubilize the coating material? |
||
|
Response 10: Thank you for your question. The coating material was dissolved in a phosphate buffer solution at pH=7.4. References Eid HM, Ali AA, Ali AMA, et al. Potential Use of Tailored Citicoline Chitosan-Coated Liposomes for Effective Wound Healing in Diabetic Rat Model. Int J Nanomedicine. 2022;17:555-575. https://doi.org/10.2147/IJN.S342504 |
||
|
Comments 11: Section 2.2.4: it has no sense. |
||
|
Response 11: Thank you for your valuable comments. This part is the method of ginsenoside Rh2 measurement and the calculation of the encapsulation rate. It has been abbreviated and changed to be combined with the section on the preparation of liposomes. |
||
|
Comments 12: RESULTS: The discussion with the literature is nonexistent. |
||
|
Response 12: Thank you for your valuable comments. We have revised the manuscript, cited the literature, and discussed the literature. |
||
|
Comments 13: Section 3.5: you must also record the spectra of physical mixtures of raw materials. |
||
|
Response 13: Thank you for your question. The spectrum of a physical mixture of raw materials that has been recorded in a spectrum is the spectrum of Empty-Lip. The black line in figure 5. |
||
|
Comments 14: In figure 6 you reported the release of the drug. Thus, you can’t talk about liposome stability in the results. Provide stability results. |
||
|
Response 14: Thank you for your valuable comments. Represented in figure 6 is the different release rate of liposomes after passing through gastrointestinal fluid, also called the leakage rate, through different environments at different times, the leakage rate of ginsenoside Rh2, the core material of the liposomes, is still very small, reflecting that there is a change for the better in the stability of liposomes in the case of the coating of polysaccharides. |
||
|
Comments 15: Section 3.7: provide the antioxidant capacities of the raw materials. |
||
|
Response 15: Thank you for your careful review and changing the picture of figure 7 in the revised manuscript to provide details of the antioxidant capacity of the raw material. |
||
|
4. Response to Comments on the Quality of English Language |
||
|
Response 1: We apologize for the errors in this manuscript and the inconvenience it caused you to read it. The manuscript has been thoroughly revised and edited by native speakers. |
||

Round 2
Reviewer 3 Report
The authors did not consider all my comments. Several studies concerning the delivery of ginsenoside Rh2 are present in the literature (6 in the last 5 years). These were not discussed by the authors. The same goes for the coating of liposomes to increase the antioxidant activity.
The results must be discussed with the literature.
The physical mixtures of the materials are still missing.
Line 106: 100 L??
Lines 107 - 112: rewrite.
Lines 183 - 183: rewrite.
Some parts are not written in a scientific manner.
Author Response
|
Response to Reviewer 3 Comments
|
||
|
1. Summary |
|
|
|
Thank you for your constructive comments on our article. We apologize for the errors in this manuscript and the inconvenience it caused you to read it. We have carefully revised the issues in the manuscript. Your helpful comments are much appreciated. These comments have been valuable and helpful in improving our article. All these comments have been carefully discussed by all the authors and the manuscript has been revised. |
||
|
2. Questions for General Evaluation |
Reviewer’s Evaluation |
Response and Revisions |
|
Does the introduction provide sufficient background and include all relevant references? |
Must be improved |
We made complete changes, which are detailed below in this document. |
|
Are all the cited references relevant to the research? |
Can be improved |
We made complete changes, which are detailed below in this document. |
|
Is the research design appropriate? |
Must be improved |
We made complete changes, which are detailed below in this document. |
|
Are the methods adequately described? |
Can be improved |
We made complete changes, which are detailed below in this document. |
|
Are the results clearly presented? |
Must be improved |
We made complete changes, which are detailed below in this document. |
|
Are the conclusions supported by the results? |
Not applicable |
We made complete changes, which are detailed below in this document. |
|
3. Point-by-point response to Comments and Suggestions for Authors |
||
|
Comments 1: The authors did not consider all my comments. Several studies concerning the delivery of ginsenoside Rh2 are present in the literature (6 in the last 5 years). These were not discussed by the authors. The same goes for the coating of liposomes to increase the antioxidant activity. |
||
|
Response 1: Thank you for your valuable comments. We have reviewed the changes. It has been found that there are many studies related to the delivery of ginsenoside Rh2. Rh2 liposomes studied by Chao Hong [17] et al were used to prolong circulation in the bloodstream, but the clinical efficacy of liposomes is still limited by the complexity of the tumor microenvironment and the insufficient accumulation at the tumor site. Minghao Sun [18] et al studied ginsenoside Rh2 microsphere hydrogels for the treatment of skin infections. However, in vitro drug release studies showed that Rh2 in the hydrogels exhibited a fast and short release pattern. The in vitro study of ginsenoside Rh2 nanoliposome formulation to enhance the antitumor effect against prostate cancer by Hadi Zare-Zardini [19] et al was used to improve the poor bioavailability of ginsenoside Rh2. (Lines 49-58) In a study, Josua Markus [20] et al found that ginsenoside Rh2 coupling provided terminal carboxyl groups to dope leaf extracts into nanocomposites to enhance Rh2 antioxidant activity against DPPH, ABTS + radicals as well as aqueous solubility. (Lines 62-64) Polysaccharide-coated quercetin liposomes, studied by Manuel Román-Aguirre [23] et al, had an enhanced effect as antioxidants on free radical scavenging activities. Clara I. Colino [24] et al. found that polysaccharide-modified liposomes protected the antioxidant activity of quercetin encapsulated in vesicles. (Lines 73-76) References [17]Hong C, Liang J, Xia J, et al. One Stone Four Birds: A Novel Liposomal Delivery System Multi-functionalized with Ginsenoside Rh2 for Tumor Targeting Therapy. Nanomicro Lett. 2020;12(1):129. Published 2020 Jun 16. [18]Sun M, Zhu C, Long J, Lu C, Pan X, Wu C. PLGA microsphere-based composite hydrogel for dual delivery of ciprofloxacin and ginsenoside Rh2 to treat Staphylococcus aureus-induced skin infections. Drug Deliv. 2020;27(1):632-641. [19]Zare-Zardini H, Alemi A, Taheri-Kafrani A, et al. Assessment of a New Ginsenoside Rh2 Nanoniosomal Formulation for Enhanced Antitumor Efficacy on Prostate Cancer: An in vitro Study. Drug Des Devel Ther. 2020;14:3315-3324. [20]Markus, J. , Mathiyalagan, R. , Kim, Y. J. , Han, Y. , Zuly Elizabeth Jiménez-Pérez, & Veronika, S. , et al. Synthesis of hyaluronic acid or o-carboxymethyl chitosan-stabilized zno–ginsenoside rh2 nanocomposites incorporated with aqueous leaf extract of dendropanax morbifera léveille: in vitro studies as potential sunscreen agents. New J. Chem. 2019; 43. [23] Román-Aguirre M, Leyva-Porras C, Cruz-Alcantar P, Aguilar-Elguézabal A, Saavedra-Leos MZ. Comparison of Polysaccharides as Coatings for Quercetin-Loaded Liposomes (QLL) and Their Effect as Antioxidants on Radical Scavenging Activity. Polymers (Basel). 2020;12(12):2793. [24] B C I C A , A D V G , A E A H ,et al.A comparative study of liposomes and chitosomes for topical quercetin antioxidant therapy. J. Drug Deliv. Sci. Technol. 2023;10(17). |
||
|
Comments 2: The results must be discussed with the literature. |
||
|
Response 2: Thank you for your valuable comments. We have revised the manuscript, cited the literature, and discussed the literature. PDI is a metric used to assess the homogeneity of particle size in suspensions. In general, the smaller the PDI, the better the particle dispersion scattering. As shown in Fig. 3A, the PDI values of Empty-Rh2, Rh2-Lip, ACP-Rh2-Lip, and CS-Rh2-Lip were elevated but remained below 0.3, which is consistent with the findings of Samuel Maritim [36] et al indicating uniform dispersion. (Lines 287-291) Josua Markus [41] et al suggested that methylene (-CH2-) is the absorption peak of polysaccharide substances in the infrared spectrum of ACP at 2884 cm-1. (Lines 329-330) Figure 6 shows that after oral digestion, Rh2-Lip is relatively well stabilized and the presence of Rh2 is rarely found. After entering the gastric fluid, the stability of liposomes varied with the change in pH value. After immersion in gastric juice for 90 min, the leakage rate of Rh2-Lip reached 50%, while the leakage rates of AC-Rh2-Lip and CS-Rh2-Lip were much lower than those of Rh2-Lip. This is consistent with the findings of Jiawei Zhang [45] et al. The reason for this may be that changes in pH disrupt the structure of the phospholipid bilayer. The neutral gastric environment and gastric electrolyte solution system altered the stability and conformation of polysaccharides, leading to partial leakage of Rh2. (Lines 346-354) Glycosidic bond breaking of ACP produces low molecular weight polysaccharides, which contain high amounts of reduced free hydroxyl groups, thus polysaccharides have antioxidant capacity. DPPH radical scavenging, hydroxyl radical scavenging, and Fe3+ reducing capacity can be used to measure the level of antioxidants [47]. (Lines 364-367) On the one hand, the thick network covering the liposome surface prevented the interaction of foreign substances with the lipid-water interface. On the other hand, the coverage of polysaccharides tightly organized the lipid acyl chains and hindered the penetration of pro-oxidant substances [48]. The CS contained more reducing hydroxyl groups, which, when combined with reactive oxygen species in the lipid peroxidation chain reaction, prevented the peroxidation process and improved the antioxidant capacity of liposomes [49]. The results suggest that polysaccharide-coated liposomes are a successful method to reduce oxidative stress in liposomes. (Lines 380-388) References [36] Maritim S, Boulas P, Lin Y. Comprehensive analysis of liposome formulation parameters and their influence on encapsulation, stability and drug release in glibenclamide liposomes. Int J Pharm. 2021;592:120051. [41] Markus J, Mathiyalagan R , Kim Y J ,et al.Synthesis of hyaluronic acid or O-carboxymethyl chitosan-stabilized ZnO–ginsenoside Rh2 nanocomposites incorporated with aqueous leaf extract of Dendropanax morbifera Léveille: in vitro studies as potential sunscreen agents. New Journal of Chemistry. 2019, 43. [45] Zhang J , Han J , Ye A ,et al.Influence of Phospholipids Structure on the Physicochemical Properties and In Vitro Digestibility of Lactoferrin-Loaded Liposomes. Food Biophysics. 2019. [47] Valgimigli L , Baschieri A , Amorati R .Antioxidant activity of nanomaterials. Journal of Materials Chemistry B, 2018:10.1039.C8TB00107C. [48] Cong L, Wang J, Lu H, Tian M, Ying R, Huang M. Influence of different anionic polysaccharide coating on the properties and delivery performance of nanoliposomes for quercetin. Food Chem. 2023;409:135270. [49] Tai K , Rappolt M , Mao L ,et al.The stabilization and release performances of curcumin-loaded liposomes coated by high and low molecular weight chitosan. Food Hydrocolloids, 2020, 99(Feb.):105355.1-105355.10. |
||
|
Comments 3: The physical mixtures of the materials are still missing. |
||
|
Response 3: Thank you for your valuable input. We apologize that we did not understand correctly what you have expressed. When you say, "the spectra of the physical mixtures of these materials are missing", does the physical mixture of these materials mentioned here refer to the mixture of soy lecithin and cholesterol? Of all the spectra we measured in Section 3.5, the spectrum measured with the Empty-Lip is that of a mixed solution of soy lecithin and cholesterol, which is the spectrum of Empty-Lip in the manuscript. Or are you expressing the spectrum of ginsenoside Rh2? If our understanding is incorrect, please point it out and we will correct it as soon as possible. |
||
|
Comments 4: Line 106: 100 L?? |
||
|
Response 4: Thank you for raising the issue. We have reworked it. We apologize for the inconvenience caused by the errors in this manuscript. |
||
|
Comments 5: Lines 107 - 112: rewrite. |
||
|
Response 5: Thank you for your valuable input. We have reworked this section. We apologize for the inconvenience caused by the errors in this manuscript. Before liposome preparation, soy lecithin was dried in a dry nitrogen blower flowing nitrogen (model: DN-12A). Soy lecithin and cholesterol were weighed and mixed with 10 ml of mixed solvent (chloroform: methanol = 6:4). The mixture was introduced into a round-bottomed flask and rotary evaporated at 50 C for 30 min to remove the residual solvent mixture and to form a uniform film on the wall of the flask. The films were then hydrated with 0.05 M phosphate buffer solution (PBS) by vortexing at 50°C for 30 min. To obtain nanoscale, the ultrasonic probe was used in an ice bath for 4 min with 5 s pulses on and 5 s pulses off. The generated liposomes were then filtered through a 0.45 um filtration membrane to obtain a liposome solution, which was transferred to a brown glass vial filled with nitrogen and stored at 4°C in a light-protected environment for further use. The encapsulation efficiency was determined using a UV-visible spectrophotometer (RM-N5000S; Ruiming Instrument Co., Qingdao, China) to observe the maximum absorbance of ginsenoside Rh2 at 544 nm. The ginsenoside Rh2 standard was dissolved in methanol to produce different concentrations of the standard solution, placed in a 50 °C water bath for 1 min to evaporate the methanol, added 0.5 ml of 8% vanillin ethanol test solution, 5 ml of 72% sulfuric acid solution, and heated by thorough mixing in a 60 °C water bath for 15 min, and then immediately used in an ice bath for 10 min and shaken well. At 544 nm, the absorbance was measured. The standard calibration curve of ginsenoside Rh2 was thus obtained (Y=0.033×X+0.0773, r2=0.999). In the above expression, Y is the absorbance at 544 nm and X is the concentration of Rh2 (μg/mL). The ginsenoside Rh2 content in the liposome supernatant was determined based on the same method described above. The encapsulation efficiency (EE) was estimated by the following equation [30]. (Lines 111-132) References [29] A, S. Y. , A, Z. W. , A, H. X. , B, H. M. A. , A, S. X. , & C, X. Y. , et al. Effect of mono- and double-layer polysaccharide surface coating on the physical stability of nanoliposomes under various environments. Colloids Surf A Physicochem Eng Asp, 2021. [30] Abbas, S. , Chang, D. , Riaz, N. , Maan, A. A. , & Afzal, M. I. . In-vitro stress stability, digestibility and bioaccessibility of curcumin-loaded polymeric nanocapsules. J Exp Nanosci, 2021; 16(1), 230-246. |
||
|
Comments 6: Lines 183 - 183: rewrite. |
||
|
Response 6: Thank you for your valuable input. We have reworked this section. We apologize for the inconvenience caused by the errors in this manuscript. The content of Rh2 in the supernatants of different samples at different stages of digestion was determined. (Lines 198-199) |
||
|
4. Response to Comments on the Quality of English Language |
||
|
Point 1: Some parts are not written in a scientific manner. |
||
|
Response 1: We apologize for any inconvenience caused by errors in this manuscript that may have caused you to read it. We have conducted a careful review and revision. |
||
